# The Impact of an Electronic Medication Management System on Medication Deviations on Admission and Discharge from Hospital

**DOI:** 10.3390/ijerph20031879

**Published:** 2023-01-19

**Authors:** Milan R. Vaghasiya, Simon K. Poon, Naren Gunja, Jonathan Penm

**Affiliations:** 1Faculty of Engineering, The University of Sydney, Camperdown, NSW 2006, Australia; 2Digital Health Solutions, Western Sydney Local Health District, North Parramatta, NSW 2151, Australia; 3Faculty of Medicine & Health, The University of Sydney, Camperdown, NSW 2006, Australia; 4Faculty of Medicine & Health, School of Pharmacy, The University of Sydney, Camperdown, NSW 2006, Australia; 5Department of Pharmacy, Prince of Wales Hospital, Randwick, NSW 2031, Australia

**Keywords:** electronic medication management system, medication safety, medication reconciliation, digital health

## Abstract

Medication errors at transition of care remain a concerning issue. In recent times, the use of integrated electronic medication management systems (EMMS) has caused a reduction in medication errors, but its effectiveness in reducing medication deviations at transition of care has not been studied in hospital-wide settings in Australia. The aim of this study is to assess medication deviations, such as omissions and mismatches, pre-EMMS and post-EMMS implementation at transition of care across a hospital. In this study, patient records were reviewed retrospectively to identify medication deviations (medication omissions and medication mismatches) at admission and discharge from hospital. A total of 400 patient records were reviewed (200 patients in the pre-EMMS and 200 patients in the post-EMMS group). Out of 400 patients, 112 in the pre-EMMS group and 134 patients in post-EMMS group met the inclusion criteria and were included in the analysis. A total of 105 out of 246 patients (42.7%) had any medication deviations on their medications. In the pre-EMMS group, 59 out of 112 (52.7%) patients had any deviations on their medications compared to 46 out of 134 patients (34.3%) from the post-EMMS group (*p* = 0.004). The proportion of patients with medication omitted from inpatient orders was 36.6% in the pre-EMMS cohort vs. 22.4% in the post-EMMS cohort (*p* = 0.014). Additionally, the proportion of patients with mismatches in medications on the inpatient charts compared to their medication history was 4.5% in the pre-EMMS group compared to 0% in the post-EMMS group (*p* = 0.019). Similarly, the proportion of patients with medications omitted from their discharge summary was 23.2% in the pre-EMMS group vs. 12.7% in the post-EMMS group (*p* = 0.03). Our study demonstrates a reduction in medication deviations after the implementation of the EMMS in hospital settings.

## 1. Introduction

Medication errors remain a concerning issue worldwide in terms of cost and harm to patients. According to the World Health Organisation (WHO), medication errors cost US $42 billion globally [1]. The Institute of Medicine in the US found that a patient could experience more than one medication error per day on average during their stay in a hospital [2]. In Australia, medications contribute to 250,000 hospital admissions annually [3]. In addition, Australian emergency departments (EDs) see 400,000 presentations with medication-related problems [3]. According to the report, 50% of the harm due to medication errors is preventable [3,4]. The literature has extensively investigated various factors that contribute to medication errors [5].

Various human factors [6], communication factors [7,8], and environmental factors [5], contribute to medication errors. Human factors such as haste, stress, distractions, long working hours, provider knowledge deficit, and inexperience [9] in medication-related tasks contribute to medication errors [7,10]. Additionally, factors such as poor communication or a lack of communication can lead to medication errors [7,8,10]. Similarly, environmental factors such as patient movement across transition of care settings, high workload, change in shift conditions, and frequent interruptions contribute to medication errors [7,8,10]. Among all the contributing factors, transition of care is one of the leading factors contributing to medication errors [8,11,12].

As a result, the World Health Organisation (WHO) made transition of care one of the three priority areas to improve medication safety in the third global patient safety challenge [12]. Transition of care includes patients moving between home, health care facilities, and residential settings (Figure 1). During patients’ stay in the hospital, there are numerous changes made to their medications [13,14]. A previous study suggests that up to 90% of patients may experience a change in their medications during their stay in the hospital [4]. Furthermore, these changes are not always understood by patients [15,16]. Additionally, healthcare providers in primary care, such as general practitioners, pharmacists, and registered nurses, are not always informed about the changes in patients’ medications post-discharge from the hospital [17,18]. As such, a transition of care can lead to medication deviations either at admission or discharge from the healthcare facility.

Medication deviations, such as omissions and discrepancies in medication lists, are routinely found during transition of care [19]. Up to 50–80% of patients have a regular medication mistakenly omitted due to incomplete documentation of patients’ medication history in Australia [20,21,22]. Upon discharge, additional errors occur, with 12–13% of discharge summaries having discrepancies in the documentation of the patient’s medication history [9]. Hence, it is important to find a solution to reduce medication deviations at transition of care.

In recent times, the use of technology has provided solutions to some of the challenges of medication management at transition of care. The use of standalone electronic tools as well as an integrated electronic medication management system (EMMS) has been shown to reduce medication errors at transition of care [23]. Additionally, previous studies showed that electronic systems have the potential to improve the completeness and accuracy of medication information [24], which can help to reduce medication deviations at transition of care.

Several studies have investigated the effect of electronic tools on medication deviations during transition of care. While some studies have evaluated the standalone electronic medication reconciliation tool, others have investigated a specific type of error (e.g., prescribing errors) after the implementation of an EMMS [25,26,27]. For example, one study found a reduction in non-intercepted medication reconciliation errors of 53% (*p* = 0.02; 95% CI 26–87%) after the implementation of the admission electronic medication reconciliation tool [25]. Similarly, another study found that the prevalence of medication errors decreased from 30% to 15% after the implementation of an electronic discharge medication reconciliation tool [26]. While one recent study evaluated medication errors at transition of care in the ICU setting [28], the effect of an integrated EMMS on medication omission and discrepancies at transition of care in the hospital-wide setting has not been studied before in Australian hospitals.

At the study hospital, the EMMS was rolled out on 28 February 2017, using a patient-centric rollout strategy, in all clinical areas except for the intensive care unit (ICU) [29,30]. The EMMS is part of a hospital-wide electronic medical record (EMR) consisting of various modules required for patient care, including observation charts, clinical documentation, and diagnostic tools. The EMMS is an end-to-end solution for medication management in the hospital setting. The system facilitates the documentation of medication history, admission reconciliation, prescription, verification, and administration of medications. The system helps staff to view all medication-related tasks in one place, and can be accessed from anywhere in the hospital.

The aim of the study was to evaluate the effect of the EMMS on medication deviations in the hospital-wide setting during transition of care. The primary objective of the study was to evaluate the effect of the EMMS on medication deviations such as omissions and mismatches before and after the implementation of the EMMS. The secondary objective of the study was to evaluate the effect of the EMMS on the completeness of documentation of medication at admission and discharge.

## 2. Method

### 2.1. Study Design

A retrospective pre-post study covering a 6-month pre-EMMS (1 July–31 December 2016) and post-EMMS (1 July–31 December 2017) implementation period was performed in a hospital-wide setting. Data for the primary outcome of medication deviations, (medication omission and medication mismatch) as well as a secondary outcome of medications documented with complete information at admission and discharge from the hospital, were collected.

### 2.2. Setting

This study was conducted at a major tertiary teaching hospital in Sydney, Australia, in February 2017. The study hospital is part of the local health district which serves a population of more than two million people. The multi-specialty hospital has a capacity of 480 acute inpatient beds and has 50,000 ED presentations annually.

### 2.3. Ethics Approval

Approval for the study was obtained from the hospital’s local human research ethics committee with reference number AU RED LNR/16/WMEAD/359.

### 2.4. Inclusion and Exclusion Criteria

Admitted patients, including paediatric patients either on regular medication at admission or discharged with medications, were included in the study. Patients admitted from 1 July 2016 to 31 December 2016 were included in the pre-EMMS group. Patients admitted from 1 July 2017 to 31 December 2017 were included in the post-EMMS group. Patients that were not admitted to the hospital and were discharged from ED, as well as patients that were admitted to the intensive care unit (ICU), were excluded from the study. Patients that were treated for day-only procedures and those seen in outpatient clinics were excluded. Patients that were discharged against medical advice (DAMA) as well as those that died in the hospital were also excluded.

### 2.5. Data Collection

A total of 200 patients were randomly selected in each group from the total admissions for the study period. Demographic variables such as age, gender, and admitting specialties were also collected from patient records. Additionally, various types of documentation, including admission notes, medications charts, inpatient notes, pharmacy notes, and discharge summaries, were reviewed to record the primary outcome of medication deviations, (1) omissions and (2) mismatches, on admission and discharge. Omissions and mismatches on admission were identified by using the most comprehensive list of medications documented on admission and comparing it with the inpatient medication charts. Inpatient records were reviewed to identify any medication changes and the rationale for the change. The reconciled list of medication made by investigators was then compared to the medications on the discharge summary to identify the omission and mismatches on discharge.

Data were also collected for the secondary outcome of (1) the number of documented medications and (2) the number of medications with complete information (name, dose, and frequency), at admission and discharge, by reviewing documentation of medication history and admission and discharge notes.

### 2.6. Definitions of Medication Deviations

The definition of medication deviations for the purpose of this study is as follows. Medication omission was counted on admission if a patient was on the medication, but the medication was not prescribed during their stay in the hospital. Additionally, medication omission on discharge was counted when a patient’s medication during their stay in the hospital was omitted from their discharge summary without any explanation. Medication mismatch on admission was identified if the matching medication was prescribed but either dose or frequency did not match the patient’s medication history. Similarly, medication mismatch on discharge was identified if the medication name was correct, but either dose or frequency were not matching. Patient records were reviewed to identify if a mismatch was intentional. If the mismatch was found to be intentional, it was not classified as a mismatch.

### 2.7. Data Analysis

The Chi-square, Fisher’s exact test, and Mann–Whitney U test were used to compare demographic and outcome variables in the pre-EMMS and post-EMMS groups. In addition, the mean and standard deviation (SD) were reported for normally distributed variables. For non-parametric variables, the median and interquartile ranges (IQRs) were reported. A two-tailed alpha of 0.05 was considered to be statistically significant. IBM SPSS statistical software version 25 was used for all the statistical analyses in this study.

## 3. Results

A total of 400 patient records were screened, with 112 patient records pre-EMMS and 134 patient records post-EMMS included for analysis, as per the inclusion and exclusion criteria. Analysis showed no significant difference in age or gender between the groups (Table 1). Patients were admitted to 20 different specialties that were combined into three main divisions. A total of 1515 medications at admission and 1863 medications at discharge were reviewed.

### 3.1. Primary Outcome: Proportion of Patients with Medication Omissions and Mismatches

A total of 105 (42.7%) out of 246 patients had any medication deviations on their medications. In the pre-EMMS group, 59 out of 112 (52.7%) patients had any deviations in their medications compared to 46 out of 134 patients (34.3%) from the post-EMMS group (*p* = 0.004) (Table 2).

The proportion of patients with medication omitted from inpatient orders was 36.6% in the pre-EMMS cohort vs. 22.4% in the post-EMMS cohort (*p* = 0.014). Additionally, the proportion of patients with mismatches in medications on the inpatient charts compared to their medication history was 4.5% in pre-EMMS compared to 0% in the post-EMMS group (*p* = 0.019). Similarly, the proportion of patients with medications omitted from the discharge summary was 23.2% in the pre-EMMS group vs. 12.7% in the post-EMMS group (*p* = 0.03). The proportion of patients with medication mismatches in the discharge summary was 3.6% in the pre-EMMS group vs. 1.5% in the post-EMMS group (*p* = 0.416).

### 3.2. Secondary Outcome: Medication with Complete Information on Admission and Discharge

As shown in Table 3, a median of four medications were documented on admission in the pre-EMMS cohort vs. a median of six medications documented in the post-EMMS cohort (*p* = 0.008). The median number of medications with complete information (name, dose, and frequency) on admission was three in pre-EMMS vs. six in the post-EMMS cohort (*p* < 0.0001). The median number of medications with name, dose, and frequency on discharge was six in the pre-EMMS group vs. seven in the post-EMMS group, with a (*p* = 0.049).

## 4. Discussion

This study showed that the integrated EMMS decreased medication deviation on admission and at discharge from the hospital. Our findings are consistent with previous studies. For example, one study found a reduction in medication reconciliation errors of 53% [25]. Similarly, another study found medication errors decreased from 30% to 15% after the implementation of the electronic admission medication reconciliation tool [26]. Reductions in medication omissions and mismatches can be a result of the clinician’s ability to select the medications from the pre-filled dropdown list of common doses and frequencies rather than manually writing a medication order. After completing the medication history, clinicians can translate the same medication into inpatient orders. This functionality of the EMMS helps clinicians to select the medication from the prefilled medication order list from the EMMS database. This potentially contributes to reducing medication omissions and mismatches after EMMS implementation.

We also found that the omission and mismatches were less between inpatient orders and discharge medications in the post-EMMS group compared to the pre-EMMS group. Our findings are consistent with previous studies. According to one study, the proportion of medication omission decreased from 16.5% to 9.1% after the implementation of the EMMS [28].

The decrease in medication deviations could be due to the results of clinician’s ability to carry out medication reconciliation with the EMMS by utilising automatic transfer of the medication orders from the patient’s medication history list or the patient’s inpatient orders into electronic discharge medications. A previous study suggested that clinician’s ability to automatically transfer medications from the EMMS to the electronic discharge summary can reduce medication errors through the elimination of the transcription process [9]. Selecting discharge medications from their medication list in the electronic system rather than manually writing medications in the discharge summary also helps to minimise the possibility of omissions of medication or mismatches in the documentation of medications (e.g., name, dose, or frequency) in a discharge summary.

This study also identified that the implementation of an EMMS improved medication completeness (e.g., medication name, dose, and frequency) during the documentation of medication history on admission and documentation of medication on discharge summaries. Our previous study supports these findings, wherein clinicians mentioned the EMMS helping them with information completeness [30].

Despite its strengths, there are some limitations of the study. While we reviewed the records retrospectively, there is a possibility that we were not able to capture medications that were verbally confirmed but not documented during transition of care. Due to limited resources, we reviewed the patient records retrospectively rather than collecting the data in real-time, e.g., collecting data while clinicians were taking medication history and documenting them in the patient records. While real-time data recording may be resource intensive, it would help to capture a higher number of medication omissions and mismatches.

## 5. Conclusions

Our study demonstrates a reduction in medication deviations after the implementation of an EMMS. Based on the retrospective data collection in the hospital setting, we found that EMMS improved specific errors related to medication deviations, such as omissions and mismatches. We attribute this improvement to EMMS’s functionality, which makes the medication process easier and simpler for clinicians by having functions that include a list of home medications and inpatient drug orders for reconciliation into the discharge medication list. While our finding shows that the EMMS improves a specific type of medication error, e.g., medication deviations, we did not study whether and how the EMMS improves the overall medication management process. Future research should investigate the effect of the EMMS on medication management processes in the hospital setting.

## Figures and Tables

**Figure 1 ijerph-20-01879-f001:**
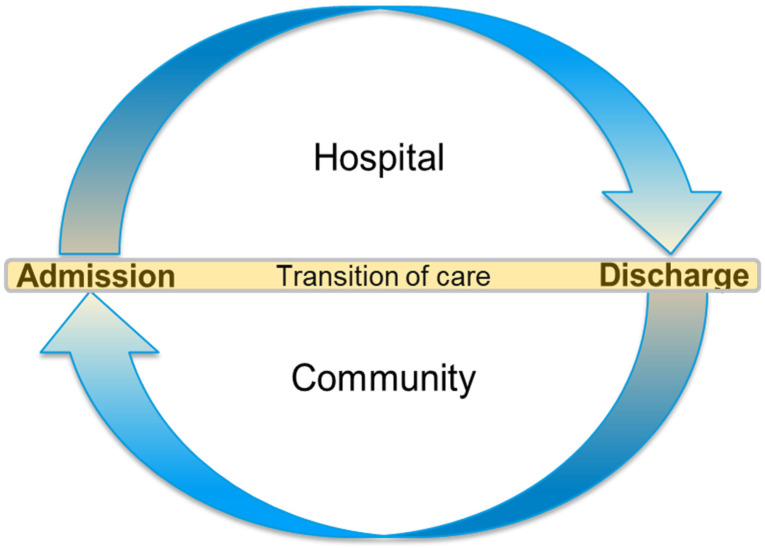
Transition of care from community to hospital and back to community.

**Table 1 ijerph-20-01879-t001:** Demographic of patients in pre-EMMS and post-EMMS in hospital-wide settings.

	Total Sample (*n* = 246)	Pre-EMMS (*n* = 112)	Post-EMMS (*n* = 134)	Statistical Significance *
Age, Mean (SD)	67.1 (18.9)	65.4 (18.9)	68.5 (18.8)	*p* = 0.467
Male, N (%)	124 (50)	52 (46)	72 (54)	*p* = 0.254
Specialty divisions				
Acute medicine, N (%)	95 (39)	42 (38)	53 (40)	*p* = 0.394
Ambulatory medicine, N (%)	109 (44)	48 (43)	61 (46)	*p* = 0.675
Surgery and anaesthetics, N (%)	42 (17)	22 (20)	20 (15)	*p* = 0.327

* Chi-square test.

**Table 2 ijerph-20-01879-t002:** Analysis of patients with a medication deviation before and after the implementation of EMMS.

Primary Outcome Variables	Total Sample (*n* = 246), N (%)	Pre-EMMS (*n* = 112), N (%)	Post-EMMS (*n* = 134), N (%)	Statistical Significance
Patients with any deviations				
Medications with deviations on admission or discharge	105 (42.7)	59 (52.7)	46 (34.3)	*p* = 0.004 *
Patients with deviation on admission				
Medications omission	71 (28.9)	41 (36.6)	30 (22.4)	*p* = 0.014 *
Medications mismatch	5 (2)	5 (4.5)	0 (0)	*p* = 0.019 **
Patients with deviations at discharge				
Medications omission	43 (17.5)	26 (23.2)	17 (12.7)	*p* = 0.03 *
Medications mismatch	6 (2.4)	4 (3.6)	2 (1.5)	*p* = 0.416 **

* Chi-square test; ** Fisher’s exact test.

**Table 3 ijerph-20-01879-t003:** Analysis of medications with complete information before and after the implementation EMMS.

Secondary Outcome Variable	Total Sample (*n* = 246), Median (IQR)	Pre- EMMS (*n* = 112), Median (IQR)	Post-EMMS (*n* = 134), Median (IQR)	Statistical Significance *
Medications per patients on admission Medications documented	5 (3–9)	4 (3–7)	6 (3–10)	*p* = 0.008
Medications with name, dose, and frequency documented	4 (1–8)	3 (0–6)	6 (2–10)	*p* < 0.001
Medications per patients at discharge Medications with name, dose, and frequency documented	7 (3–10)	6 (2–9)	7 (4–11)	*p* = 0.049

* Mann–Whitney test.

## Data Availability

The dataset used for this study is available from the corresponding author upon a reasonable request.

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
