# Peer review of "The Impact of an Electronic Medication Management System on Medication Deviations on Admission and Discharge from Hospital"

_ijerph, 2023, doi:10.3390/ijerph20031879_

Round 1

Reviewer 1 Report

This article addresses the positive results of the use of electronic medication management system at transition care. However, I would like to make some suggestions.

Abstract:

The authors wrote: “In this study, the patient records were reviewed retrospectively to identify medication deviations (medication omission and medication mismatch) at admission and discharge from the hospital”

·       -From... (date) until .... (date) were the patient records included?

·       -What's the hospital? (a tertiary teaching hospital in Sidney)

 Introduction

The EEM system description included in the section "Method" could be mentioned in the section "Introduction", before the objectives of the study.

Method

2.5. Inclusion and exclusion criteria

The authors wrote: “Admitted patients either on regular medication on admission…”

·       Did the authors include pediatric patients?

 Results

·       - Did the authors identify any differences in medication deviation between specialties?

-I ask the authors to have attention in table 2:

 105 +71+ 5+ 43 + 6 = 230 (table: n=246) ??

59 + 41 + 5 +26+ 4= 135 (table: n=112) ??

46 + 30+ 0+ 17+2 = 95 (table: n=134)   ??

Reviewer 2 Report

In this paper, Vaghasiya et al. assess medication deviations pre-Electronic Medication Management Systems (EMMS) and post-EMMS implementation during the transition of care. Based on retrospective data collection, the authors find that EMMS improved specific errors related to medication deviations, such as omissions and mismatches. 

This paper is clear and well-written. Language is appropriate. I have only a minor point: can the authors explain the difference between “acute medicine”, “ambulatory medicine” and “surgery and anaesthetics”. These items are listed in Table 1.

Reviewer 3 Report

The impact of Electronic Medication Management System on medication deviations on admission and discharge from hospital is an interesting and well-written paper. The manuscript has the potential for publication consideration but before that, some issues must be solved:

Line 101: exact dates of the six months should be added.

The authors should explain why did they focus only on the six months if EMMS was implemented in hospitals in 2017?

Table 1: Results of statistical analysis should be added (e.g. Chi’s square test) and only average and p-value. Legend is missing… what are the numbers in brackets? what was significance the significance limit? Why only p-value for acute medicine and not also for ambulatory and surgery?

Correct P-value to p-value.

The same for Table 2 and Table 3

Explain “5 (3, 9)” What are 3 and 9?

Round 2

Reviewer 3 Report

No further comments.